# Exome Sequencing Identifies Genetic Variants Associated with Extreme Manifestations of the Cardiovascular Phenotype in Marfan Syndrome

**DOI:** 10.3390/genes13061027

**Published:** 2022-06-08

**Authors:** Yanireth Jimenez, Cesar Paulsen, Eduardo Turner, Sebastian Iturra, Oscar Cuevas, Guillermo Lay-son, Gabriela M. Repetto, Marcelo Rojas, Juan F. Calderon

**Affiliations:** 1Doctorado en Ciencias e Innovación en Medicina, Facultad de Medicina Clínica Alemana Universidad del Desarrollo, Santiago 8320000, Chile; yjimenezb@udd.cl (Y.J.); marojash@udd.cl (M.R.); 2Servicio de Cirugía Cardiovascular, Instituto Nacional del Tórax, Santiago 7500808, Chile; cesar.paulsen@gmail.com (C.P.); eturner@manquehue.net (E.T.); sebastian.iturra@gmail.com (S.I.); ocuevas@alemana.cl (O.C.); 3Departamento de Cirugía Cardiovascular, Clínica Alemana, Universidad del Desarrollo, Santiago 8320000, Chile; 4Unidad de Genética, División de Pediatría, Facultad de Medicina, Pontificia Universidad Católica de Chile, Santiago 8331150, Chile; grlayson@gmail.com; 5Programa de Enfermedades Poco Frecuentes, Centro de Genética y Genómica, Instituto de Ciencias e Innovación en Medicina, Facultad de Medicina Clínica Alemana Universidad del Desarrollo, Santiago 8320000, Chile; grepetto@udd.cl; 6Centro de Genética y Genómica, Instituto de Ciencias e Innovación en Medicina, Facultad de Medicina Clínica Alemana Universidad del Desarrollo, Santiago 8320000, Chile; 7Research Center for the Development of Novel Therapeutic Alternatives for Alcohol Use Disorders, Santiago 8320000, Chile

**Keywords:** Marfan syndrome, aortic aneurysm, genetic modifiers, exome sequencing

## Abstract

Marfan Syndrome (MFS) is an autosomal dominant condition caused by variants in the fibrillin-1 (*FBN1*) gene. Cardinal features of MFS include ectopia lentis (EL), musculoskeletal features and aortic root aneurysm and dissection. Although dissection of the ascending aorta is the main cause of mortality in MFS, the clinical course differs considerably in age of onset and severity, even among individuals who share the same causative variant, suggesting the existence of additional genetic variants that modify the severity of the cardiovascular phenotype in MFS. We recruited MFS patients and classified them into severe (*n* = 8) or mild aortic phenotype (*n* = 14) according to age of presentation of the first aorta-related incident. We used Exome Sequencing to identify the genetic variants associated with the severity of aortic manifestations and we performed linkage analysis where suitable. We found five genes associated with severe aortic phenotype and three genes that could be protective for this phenotype in MFS. These genes regulate components of the extracellular matrix, TGFβ pathway and other signaling pathways that are involved in the maintenance of the ECM or angiogenesis. Further studies will be required to understand the functional effect of these variants and explore novel, personalized risk management and, potentially, therapies for these patients.

## 1. Introduction

Marfan syndrome (MFS, OMIM #154700) is a rare connective tissue disorder with a reported incidence of 1 in 3000 to 5000 individuals [1]. MFS has autosomal dominant inheritance and is caused by variants in *FBN1*, the gene encoding for fibrillin-1 an extracellular matrix (ECM) protein that is the major structural component of microfibrils [2,3]. The clinical features of MFS include aortic root aneurysm and/or dissection, ectopia lentis and overgrowth of large bones, among others [4].

Clinical manifestations of MFS are highly variable with significant phenotypic heterogeneity even among individuals harboring the same pathogenic variant in *FBN1*, including intrafamilial variability [5,6]. Prognosis in MFS is related to the life-threatening complications of the thoracic aortic kind as dissection of aorta is the main cause of mortality. The clinical course of cardiovascular manifestations in MFS differs considerably in age of onset and severity [7], although the reasons of this variability remain largely unexplained. However, some studies have shown that different pathogenic variants in *FBN1* produce among other effects, variations in its expression and suggest that different MFS-causing genotypes could modify the severity of the disease [8,9,10,11]. In contrast, other studies on mice models of MFS have identified other loci that modulate the cardiovascular phenotype of Marfan syndrome [12,13,14].

The main mechanisms by which aortic aneurysms (AAs) occur in MFS are largely unknown. However, there is significant evidence that aneurysms are a consequence of changes in the ECM integrity of the aortic wall and that these would be mediated by perturbations in the transforming growth factor-β (TGFβ) signaling pathway [15]. This perturbation occurs because fibrillin-1 is a regulator of the bioavailability of TGFβ; therefore, a defective fibrillin-1 can no longer sequester TGFβ increasing its activity levels [16], which triggers inflammation, fibrosis, and activation of metalloproteinases MMP-2 and MMP-9, leading to significant loss of vascular smooth muscle Cells (VSMCs) [15]. These factors, together with the decrease in collagen content of ECM, reduce aortic structural integrity and lead to dilation or aneurysm in MFS [17]. In fact, it has been shown in a mouse model of MFS that inhibition of TGF-β signaling prevents the development of cardiovascular phenotypes, and this is independent of the causal variant in the *FBN1* gene [18].

Knowledge of the effects of genetic variants elsewhere outside of *FBN1* on the frequency and progression of the cardiovascular phenotype in patients with MFS is very limited; therefore, deepening our understanding of this phenomenon is very important to improve the clinical management/risk stratification of patients.

In the present study, we evaluated genetic variants that could modify the severity of the clinical course of aortic aneurysm or dissection in patients with MFS using exome sequencing and linkage analysis.

## 2. Materials and Methods

Chilean patients of all ages with a confirmed diagnosis of MFS were identified and recruited for this study. Among those, we selected 5 families with both affected and unaffected members as well as 5 non-familial cases with MFS. We focused on the most life-threatening aspect of MFS: aortic aneurysm and dissection. In order to test the hypothesis that variants in modifying genes may mitigate or exacerbate the severity of the aortic phenotype, we classified our subjects in “mild” and “severe” aortic phenotype. MFS patients were clinically diagnosed according to the criteria published in the revised version of the Ghent criteria for MFS [19]. Since these guidelines define the “archetypical” phenotypes of MFS, we stratified patients that could be placed in the extremes of these parameters. Classification criteria used in this study are summarized in Appendix A. Patients were classified as mild or severe aortic phenotype if they fulfilled at least one criterion in each respective category.

Institutional Ethics Committee approved the study registry protocol at both recruitment sites (Comité Ético Científico, Facultad de Medicina Clinica Alemana Universidad del Desarrollo and Servicio de Salud Metropolitano Oriente, Department of Health) and written informed consent was obtained from all participants.

### 2.1. Exome Sequencing and Variant Discovery

We used Exome Sequencing to identify genetic variations that may be associated with the severity of the cardiovascular phenotype in MFS. Patients with MFS belonging to the 5 families studied, their informative healthy relatives and all non-familial cases with MFS were sequenced for further analysis.

Exome Sequencing was performed at BGI using their proprietary DNBseq™ NGS Platform as well as Agilent V6 + UTR library kit with depth of coverage 100×. Variant analysis was performed following the best practice recommendations of the Genome Analysis Toolkit (GATK) [20]. Briefly, raw sequences were aligned to the human reference genome (version hg19) using the Burrows–Wheeler Aligner (BWA) [21] and were processed by Picard (Available online: http://broadinstitute.github.io/picard (accessed on 12 April 2022) [22] to remove PCR duplicates. Next, local realignment around indels and base quality score recalibration were done using GATK IndelRealigner and BaseRecalibrator respectively [23]. Variant calling was performed using HaplotypeCaller. Raw variants quality scores were recalibrated using GATK VariantRecalibrator and ApplyRecalibration to generate a Variant Quality Score Logarithm of Odds (VQSLOD) [23]. Before proceeding to use VAAST to identify modifier variants, we used ANNOVAR on the filtered VCF files to confirm the presence of an MFS-causing variant in FBN1 in all the included patients.

### 2.2. VAAST Analysis

We performed variant prioritization analysis using the Variant Annotation, Analysis, and Search Tool (VAAST 2.0) package [24]. VAAST uses an extended Composite Likelihood Ratio Test (CLRT) to determine a severity score for genomic variants [25]. CLRT operates under the assumption that the frequency of a variant or variant group is the same in the control population (denominated background genomes in the software usage) and case population (denominated target genomes in the software usage), while the alternative model allows these two frequencies to differ. Under a binomial distribution, the likelihood for both models can be calculated based on observed allele frequencies in the control and case datasets. VAAST provides several ways to explore genomic datasets for families and individuals and it has been successfully used to identify both causal genes and modifier genes for several genetic disorders. In this regard, VAAST has made it possible to identify candidate modifier genes for different phenotypes, including immune function in 22q11.2 deletion syndrome and chronic obstructive pulmonary disease (COPD) susceptibility [26,27]. The variants were annotated for their functional impact using Variant Annotation Tool (VAT). The annotated variants in all patients with mild and severe phenotype were combined using Variant Selection Tool (VST). VAT analysis was performed in each of the families that had mild and severe cases (families 1, 3 and 5) and in a combination that includes all mild and all severe familial and non-familial cases. Phenotype/putative modifying genes and variants for MFS were analyzed using VAAST to prioritize candidate genes under both dominant and recessive modes of inheritance.

In this work we generated Exome Sequencing data from small families and also from non-familial cases, and consequently, we analyzed the data in two ways: (1) the familial approach, in which we analyzed the families that had both members affected with severe phenotype and affected with mild phenotype; and (2) the case-control approach, in which we used the exome data of all patients (familial and non-familial) classified as either severe or mild phenotype and performed a case-control analysis using VAAST [24].

In the first approach (Familial), we performed VAAST analysis in families 1, 3 and 5. Because no data of any affected individual from generation I was available that could link affected patients from generation II by ancestry, we could not do linkage analysis for the entire family 1. However, family 1 was partitioned in two distinct nuclei and analyzed considering mild and severe patients with the possibility of identifying alleles shared by IBD (identical by descent) segments, as shown in Figure 1. Therefore, we analyzed subfamily 1.1 (composed by II-1, II-2 and III-1 individuals) and subfamily 1.2 (composed by II-3, II-4, III-6 and III-8 individuals).

In the case-control approach, Exome Sequencing data of all MFS patients classified as severe and all MFS patients classified as mild were analyzed. Because we could not estimate the effect size of the putative modifier variants/genes, i.e., big effect size under a “monogenic-like” model or a small effect size under a “polygenic-like” model [28], we performed our analyses using the “incomplete penetrance” parameter. This allowed to identify both genes with a strong effect present in all individuals with a given shared phenotype but also genes with a lower effect that may be present only in some of the individuals with the shared phenotype to be studied. In keeping with the approach, we performed the VAAST analysis assuming incomplete penetrance and under two different schemes: (a) the autosomal dominant mode of inheritance that allowed selecting genes that had at least one copy of the putative variant; and (b) the autosomal recessive mode of inheritance that selected genes with two alleles that differed from the reference for any given coordinate (this could be identical alleles or compound heterozygotes). Additionally, we allowed for the presence of locus heterogeneity in order to perform an unbiased search across the entire exome.

### 2.3. Variant Prioritization

We filtered for and selected genes that met any of the following criteria: (1) Gene Ontology (GO) for biological processes distinctively associated with pathophysiology of MFS; (2) genes whose products interact with proteins that play a role in the homeostasis of the ECM or in the TGFβ signaling pathway (selected using STRING [29]); or (3) genes previously associated with cardiovascular phenotypes as annotated in the literature. Finally, the variants of candidate genes obtained with the VAAST analysis of the families with individuals with severe and mild phenotypes were searched for in each of the non-familial mild and severe patients and those genes with variants that did not show a clear segregation with the mild or severe phenotype were discarded. To identify candidate genes from the Exome Sequencing dataset, we selected both rare variants with presumably large effect sizes with a deleterious or harmful impact according to the SIFT calculation [30], as well as non-deleterious variants that, in association with the causal variant of MFS (*FBN1*), could explain changes in the function of a molecular pathway associated with the pathophysiology of MFS.

Raw sequencing data will be available under Bio Project accession number PRJNA795044 upon publication of this manuscript.

## 3. Results

### 3.1. Subjects

According to the age of presentation of the first cardiovascular manifestation and/or accident related to the aorta, we classified 8 patients as severe phenotype and 14 as mild (Appendix A). WES was performed on these 22 MFS subjects along with 3 informative healthy relatives (Figure 1). No significant differences were observed in relation to the distribution of gender and age between the groups of mild and severe patients (*p*-value = 0.0743). A total of 50% of severe cases already required aortic surgery (4/8) while approximately 30% of mild cases have required this surgery (4/14). Our first analysis was to confirm the presence of an MFS-causing variant on the *FBN1* gene in all recruited patients, shown in Table 1.

**Table 1 genes-13-01027-t001:** Identification of MFS-causing, pathogenic variants in *FBN1* in study subjects.

Family or Sporadic Cases	Patient ID	*FBN1* Variants HGVS Nomenclature (NM_000138.5)	Variant Type	Clinical Significance (SIFT)	Protein Functional Domain	Aortic Phenotype Classification
Fam 1	CAS-01-001	c.7339G>A	Missense	Pathogenic	EGF-like calcium-binding domain	Severe
	CAS-01-002					Mild
	CAS-01-003					Mild
	CAS-01-007					Mild
	CAS-01-005					Severe
Fam 2	CAS-01-019	c.1090C>T	Nonsense (stop gained)	Pathogenic	TB domain	Mild
	CAS-01-020					Mild
	CAS-01-021					Mild
	CAS-01-022					Mild
	CAS-01-023					Mild
Fam 3	CAS-01-024	c.7180C>T	Nonsense (stop gained)	Pathogenic	-	Mild
	CAS-01-026					Severe
Fam 4	CAS-01-035	c.7204+1G>A	splice region	Likely_pathogenic	-	Mild
	CAS-01-036					Mild
Fam 5	CAS-01-045	c.4196_4197insA	frameshift insertion	Pathogenic	EGF-like calcium-binding domain	Severe
	CAS-01-046					Mild
	CAS-01-048					Mild
Sporadic 1	CAS-01-016	c.8562delC	frameshift deletion	Pathogenic	-	Severe
Sporadic 2	CAS-01-027	c.1090C>T	Nonsense	Pathogenic	TB domain	Severe
Sporadic 3	CAS-01-031	c.5788+5G>T	splice region	Pathogenic	-	Severe
Sporadic 4	CAS-01-043	c.A2673G	Regulatory region	Likely_pathogenic	TB domain	Severe
Sporadic 5	CAS-01-044	c.236dupA	frameshift insertion	Pathogenic	N-terminal domain	Mild

### 3.2. VAAST Analysis

Table 2 shows the number of genes obtained after analysis with VAAST in the different comparisons. From the VAAST output, several hundred genes had a CLRT-derived *p* value < 0.05 in each analysis setup after performing the Bonferroni correction for multiple comparisons. The complete list of genes with statistical significance identified after performing each of the analyzes using VAAST is shown in Appendix A.

The primary list of genes obtained from the different analyses mentioned above is large, and, in some analyses, it is composed of hundreds of genes. In order to situate these candidate genes in a biological context, several avenues were explored. Considering all the selection criteria established in this study, we selected eight genes that met those criteria. A graphical description of the selection process is shown in Appendix A. Among these, variants in five genes were associated with the severe aortic phenotype and three with the mild aortic phenotype (Table 3). These genes were selected both due to the VAAST statistical criteria and also because most of them are expressed in the aorta according to GTEx consortium data [31].

### 3.3. Variant Prioritization and Candidate Modifier Genes

#### 3.3.1. Candidate Genes Associated with the Severe Cardiovascular Phenotype in MFS

We identified five candidate genes associated with a severe aortic phenotype: the first is *PPARD* (Peroxisome Proliferator-Activated Receptor δ). We found a missense variant (c.140G>A) with an autosomal dominant model of inheritance (Table 4) in exon 4 that changes the amino acid arginine by the amino acid glutamine at the conserved position 47 of the protein. This change was observed in a patient with severe aortic phenotype (III1) in family 1.1 but was absent in patients with mild aortic phenotype in the same family or any other patients with a mild aortic phenotype analyzed in this study (Figure 1). This variant is located in an intrinsically disordered protein (IDP) domain that plays important signaling and regulatory functions and has been predicted to be involved in diseases [32]. It is known that TGF-β1 is a target gene for PPARδ in VSMCs [33] and that PPARδ plays a potential role in the modulation of ECM homeostasis through the regulation of the synthesis and degradation of extracellular matrix components through the transforming growth factor-β1 and its effector Smad3 [34]. In this regard, Hyo Jung Kim et al. reported that the activation of PPARδ increases the expression of collagen types I and III, fibronectin, elastin and TIMP-3 (tissue inhibitor of metalloproteinases 3). Furthermore, it decreases the apoptotic cell death induced by oxidized low-density lipoproteins and elastase in aortic VSMCs [34]. On the other hand, in healthy blood vessels, VSMCs reside within the lamina media and remain inactive. There is evidence that the proliferation and abnormal migration of VSMCs are a common event in the pathophysiology of many vascular diseases, where these cells proliferate and migrate from the media to the intima, leading to hyperplasia and vascular stenosis [35,36,37]. PPARδ activation represents protection against endothelial damage and dysfunction, as well as vascular cell proliferation through various mechanisms [38]. In this regard, studies suggest that PPARδ significantly inhibits PDGF-induced proliferation in VSMCs by repressing expression of cyclin D1, cyclin D3, CDK2 and CDK4 [39]. In summary, PPARδ plays a prominent role in maintaining vascular integrity via suppressing VSMCs proliferation, inhibiting VSMCs migration with preservation of extracellular matrix (ECM), and also inhibiting apoptosis and senescence of VSMCs by upregulating antioxidant genes and suppressing inflammation. We believe that the effect that PPARδ exerts in protecting the integrity of the vascular structure of the aorta could be slightly counteracting the damaging effect produced by a defective fibrillin 1 present in patients with MFS, and that in keeping with this hypothesis, missense variants in the *PPARD* gene could be decreasing its protective function of the vascular structure and promoting the severe aortic phenotype in these patients.

When a recessive inheritance model was used to perform the analysis of family 1.1 in VAAST, a second missense variant in the *FBN1* gene (c.1240C>A) (Table 4), additional to the causative variant of MFS, was found. In this case, the patient III1 of the pedigree of family 1 (Figure 1) was classified as “severe” and has one variant in *FBN1* that she shares with her mother (MFS-causing variant) and a second variant in *FBN1* that is shared with her father (Non-pathogenic variant). This variant substitutes the conserved, non-polar amino acid Proline by the polar amino acid Threonine in position 414 of the protein, and it is located in a domain where it has been predicted that folding that contributes to tertiary structure of this protein occurs [40]. It is well known that variants that produce changes from a non-polar to a polar group significantly modify the characteristics of a protein and therefore could modify its function [41]. It has been studied previously that variants in *FBN1* cause variation in the expression levels of the gene and, in light of this, this second *FBN1* variant in this patient could be considered as a modifier gene that favors the severity of MFS [13,14,42].

Interestingly, another candidate gene that could modulate the cardiovascular phenotype was prominent in our results. Two missense variants in *JAG1* (c.3355G>C and c.3354T>A) (Table 4) segregated with severe aortic phenotype in MFS in family 3 under a recessive inheritance model. These variants alter a well-conserved amino acid (Gln1118Pro) in an evolutionarily conserved cytoplasmic domain of JAG1. The *JAG1* gene encodes a protein called Jagged-1, which interacts with the Notch signaling pathway. Jagged-1 connects with Notch receptors (Notch1 to Notch4), resulting in the release of the Notch Intracellular Domain (NICD) by proteolytic cleavage and subsequent translocation of NICD to the nucleus, where it interacts with the Recombinant Binding Protein Suppressor (RBPJ) and other transcription factors to regulate the activation and suppression of target genes that initiate a signaling cascade involved in vascular wall homeostasis and remodeling [43,44,45]. Jagged-1 normally activates Notch in VSMCs in the lamina media of the aorta, spreading a signaling pulse that is crucial to induce the differentiation of the VSMC layer towards the contractile phenotype to which VSMCs must return after undergoing a phenotypic switch to the synthetic phenotype in order to promote remodeling and, ultimately, to maintain vascular homeostasis in response to hemodynamic alterations [44,45,46]. We hypothesize that missense variants found in *JAG1* could downregulate the Jagged1-Notch signaling and therefore impede proper restoration of vascular wall homeostasis.

Another candidate gene was *MAP3K1*. We identified variants associated with the severe phenotype in this gene both in the familial approach (family 3) and in the non-familial, case control one (Table 4). Thus, we identified three missense variants (c.14C>G, c.2587G>T and c.362G>A located in conserved amino acids of the MAP3K1 protein Ala5, Val863 and Gly121, respectively). Interestingly, another patient with a severe aortic phenotype presented a deletion (c.1504_1505+101del) that encompasses the last two nucleotides of exon 1 and a major portion of intron 1. We hypothesize that this generates an aberrant mRNA that contains part of intron 1 and that incorporates a premature termination codon (PTC) that would undergo nonsense mediated decay (NMD) and render this allele as non-functional.

*MAP3K1* encodes for the Mitogen-activated protein kinase kinase kinase 1, which normally works inhibiting ECM cell shedding-induced apoptosis (anoikis). One consequence of ECM degradation in the pathophysiology of MFS is the increase in MMPs [47], which leads to degradation of fibronectin [48] and detachment of VSMCs and their subsequent apoptosis [49]. This apoptosis process induced by detachment of cells (anoikis) is controlled by the effect of the MAP3K1 protein and reduces cell death by activating pro-survival targets [50,51].

*PTPRJ* is another gene that showed variants that segregated in patients with severe phenotype but not in patients with mild phenotype (frameshift variant c.2786_2786+73del, in-frame deletion c.66_68del and missense variant c.2182G>A that produces the Glu728Lys substitution in the protein). Two of these variants (c.66_68del and c.2182G>A) were predicted to have a damaging or deleterious effect that modifies the structure and function of the protein by SIFT (Table 4). c.2786_2786+73del frameshift variant is a 74 bp deletion that includes the last nucleotide of exon 13 and part of the next intron; therefore, there is a loss of the 5’ splicing donor site that generates an aberrant mRNA with 14 additional amino acids generated from intronic sequence and also a premature termination codon (PTC). The variants c.2786_2786+73del and c.2182G>A are located on the non-cytoplasmatic Fibronectin type III domain, which is an evolutionary conserved protein domain. We conclude that these variants could then have a major impact on the structure and function of the protein. The in-frame deletion variant (c.66_68del) is a 3 bp deletion that causes a loss of a single leucine residue in the protein localized in the signal peptides region of PTPRJ and its effect on translational suppression of PRPRJ could be assessed [52].

*PTPRJ* or protein receptor tyrosine phosphatase type J (also named CD148 or DEP-1) is expressed in vascular endothelial cells. CD148 has the capacity to modulate cell proliferation, cell migration and is involved in vascular development through negative regulation of VEGF receptor 2 (VEGFR2) signaling [53,54]. Variants in *PTPRJ* have been associated with coarctation of the aorta [55]. A recent study reported that *PTPRJ* is upregulated in the aortic tissue of patients with thoracic aortic aneurysm (TAA) and could predict the presentation of TAA [56]. Although these results are still not very clear, suggest that PTPRJ could be exerting some function for maintenance of homeostasis of the aorta. Loss-of-function of *PTPRJ* due to the presence of gene variants could be associated with TAA produced by the uncontrolled activation of the TGFβ signaling pathway in patients with variants in *FBN1*.

#### 3.3.2. Candidate Genes Associated with the Mild Cardiovascular Phenotype in MFS

Our VAAST analysis showed three more candidate genes (*KIAA1462*, *TNFSF18*, and *TGFBR3L*) (Table 4) that segregated with mild aortic phenotype, suggesting that they could be mitigating the severity of cardiovascular phenotype in MFS, rendering variants with a protective effect. Two of these, *KIAA1462* and *TNFSF18*, are involved in degradation of the ECM through the activation of metalloproteinases. For *KIAA1462*, we identified three variants associated with the mild phenotype. A frameshift variant (c.2574_2576del) modifies the reading frame and generates the loss of a serine in the amino acid sequence. We also identified an in-frame variant (c.2577_2578insACTGCTGCT), which produced an insertion of the amino acids Thr-Ala-Ala in the protein. Finally, we found a missense variant (c.424C>T) that changes the conserved amino acid Ala142 to Thr. KIAA1462, also known as *JCAD* (Junctional Cadherin 5 Associated), is a cell–cell binding protein expressed primarily in endothelial cells [57] and regulates pathological angiogenesis, rather than developmental angiogenesis [58].

Recent studies have shown that the depletion of KIAA1462 decreased the expression of genes such as *VCAM1* [59], which is involved in the pathogenesis of ascending aortic aneurysm through increasing the activity of MMP-2 and MMP-9 that promotes aortic aneurysm progression [60]. Jones et al. described a model for the effect of JCAD in conjunction with the Hippo signaling pathway in coronary artery disease (CAD). They suggest that JCAD acts downstream of RhoA to inhibit LATS1/2, and, in turn, there is a decrease in phosphorylation of YAP/TAZ [61]. It has been reported that in dissection and aneurysm of the aorta, downregulation of YAP/TAZ promotes the apoptosis of VSMCs [62], while Liu et al showed that downregulation of YAP/TAZ induces apoptosis of VSMCs and aortic dissection and that overexpression of YAP/TAZ is able to reverse this effect [63]. We believe that it is probable that variants in the *JCAD* gene could decrease its expression and, in turn, generate an increase in YAP/TAZ that, as mentioned above, could produce a protective effect for the appearance of aortic dissection. On the other hand, we identified two missense variants (c.361G>A and c.451A>G) in the gene *TNFSF18* that change amino acids that are largely conserved: Gly121Ser and Asn151Asp, respectively. It is worth noting that *TNFSF18* was highly prioritized on our VAAST analysis despite not being expressed in aortic tissue. However paradoxical, this only highlights the necessity of further investigating the biological role of this gene in aortic homeostasis.

Additionally, we found a frameshift insertion (c.455_456insTTG) that incorporates a premature termination codon (PTC) and generates a shorter, aberrant mRNA. All the variants found in this gene are in an extracellular domain and using pDomTHREADER (recognition algorithm) was recognized how folk domain; therefore, these variants could produce a conformational change of the protein. TNFSF18 (member 18 of the tumor necrosis factor ligand superfamily) activates the expression of MMP-2 and MMP-9, but by phosphorylation of p-STAT1, and positively regulates the expression of VCAM1 and ICAM1 [64].

*TGFBR3L* is an important paralog of the *TGFBR3* gene that encodes for a coreceptor for TGF-β and has a role in the regulation of the transforming growth factor receptor beta signaling pathway superfamily by promoting the binding of TGFβ2 to the TGFβR1/TGFβR2 receptor complex [65]. One of the functions of TGFβR3 is to serve as a presenter of the TGFβ2 ligand to the TGFβR2 receptor that will then form a heterodimeric TGFβR1-TGFβR2 complex, and when this is phosphorylated, the TGF-β signaling cascade is initiated [65,66]. In this sense, it has been shown that activation of TGFBR3 is greater in the presence of variants in *FBN1* and is involved in the over-activation observed in patients with MFS [67]. We identified an in-frame variant in *TGFBR3L* (c.418_420del) that generates a loss of a codon in the mRNA that encodes for amino acid Pro140 and that is located in the zona pellucida (ZP) [68] domain of TGFBR3. The C- region of this protein, where this amino acid is located, regulates both the polymerization of extracellular matrix proteins and the recognition of TGFB by theTGFBR3 receptor [69], suggesting that loss of Pro140 could impair the activation of TGFBR3 and therefore downregulate the uncontrolled activation of the TGFB signaling pathway, thus protecting carriers against a severe aortic phenotype.

#### 3.3.3. In Silico Analysis of the Effect of Variants in Protein Structure

When crystallization data was available, we decided to perform 3D modeling of proteins to assess whether the variant identified in MFS patients with extreme aortic phenotype had a major effect in structure that could lead us to hypothesize a functional consequence of the genetic variant. We were able to retrieve PDB data for MAP3K1, PTPRJ, TNFSF18 and TGFBR3L that we then modeled using Swiss-PDBViewer [70]. We then generated molecular visualization images using the PyMOL Molecular Graphics System, Version 2.0 Schrödinger, LLC. The 3D protein structures of reference sequences and sequences with the identified variants are shown in Appendix A.

## 4. Discussion

Our results show that there are variants in genes that regulate components of the extracellular matrix, TGFβ pathway and in genes that function in other signaling pathways that are involved in the maintenance of the ECM or angiogenesis and that could modify the cardiovascular phenotype in MFS. We found five candidate genes from the VAAST analysis associated with severe aortic phenotype in MFS patients and three genes that could be protective for this phenotype.

In the present study, we recruited patients with a clinical diagnosis of MFS in agreement with the Ghent criteria for MFS. Based on these, we stratified all MFS patients as “severe” or “mild” depending on the age at which they underwent their first surgical repair of the aorta or the age at which they presented their first event of aortic dissection. With this approach, we sought to differentiate patients according to the point in time at which their aortic wall failed to sustain a normal architecture as a measure of how severe perturbations in the underlying molecular mechanisms present in each of our study participants.

Fibrillin 1 plays a pivotal role in regulating the bioavailability of TGF-β [71,72,73], and MFS-causing variants in this gene considerably alter ECM homeostasis by disrupting fibrillin 1 ability to maintain TGF-β sequestered and inactive. This, in turn, pervasively increases the levels of TGF-β ligand in the extracellular space [73]. TGF-β is a molecule that regulates many biological processes, including apoptosis, collagen production and remodeling of the extracellular matrix (ECM), among others [74,75]. Therefore, dysregulation of TGF-β interferes with maintenance of homeostasis of the aortic wall and its effect depends on the bioavailability of TGF-β in the aortic tissue, but also on other factors that could play a key role in patients with MFS and that have not been fully described. Discovery and characterization of these other factors could be key to explain differences in aortic phenotype in patients with MFS. In this sense, identification of genes (and gene products) that could modulate active TGF-β levels should be a top priority to understand the variable expressivity observed in the cardiovascular phenotype of MFS patients. Therefore, our search for modifier genes was aimed at identifying elements that either increase TGF-β activation or that interfere in the degradation process of the ECM in such a manner that favor the appearance of an aortic phenotype. Alternatively, we aimed to identify genes that could downregulate the uncontrolled activation of TGF-β, which, in keeping with our hypothesis, would render protective to the normal architecture of the aorta.

This study pinpoints several genes that play a role in the complex architecture of aortic homeostasis and genetic variants that may significantly affect their function to the extent of possibly causing a major shift in the progression of aortic phenotype in these patients. In an attempt to have a first glimpse of the mechanisms by which the identified variants cause the extreme aortic phenotypes observed in MFS patients, we modeled the proteins with the amino acid substitutions and insertions/deletions where suitable to assess whether overt changes in protein structure could be observed. For most proteins modeled, modest changes are observed. However, this observation must be correlated with complementary studies such as eQTL analysis and functional testing of the variants reported here in appropriate in vitro and in vivo systems in order to deepen our understanding of this biological phenomenon.

## 5. Conclusions

Patients with MFS present a high variability in the severity of the aortic phenotype. Understanding this phenotypic variability would allow predicting clinical outcomes in these patients, taking more effective measures for proper clinical management and avoidance of life risk situations. In this study, we proposed a novel strategy to stratify MFS patients into mild or severe aortic phenotype, which made it possible to estimate risk profiles through the identification of variants associated with these extreme phenotypes.

Although Exome Sequencing has become an important tool for the discovery of genetic variants, its use has some limitations in relation to the loss of important information in genomic regions with insufficient coverage (false negatives). The search for both pathogenic and modifier variants with Exome Sequencing data is therefore done at risk of the loss of information in non-coding regions of the human genome. We hope to further our studies using Genome Sequencing data in the future to overcome this liability.

Finally, future studies are needed to determine the molecular mechanisms by which these genetic variants could act to modify the severity of the aortic phenotype in these patients, and whether these genes are involved in the development of aortic aneurysm using in vitro and in vivo experiment models.

## Figures and Tables

**Figure 1 genes-13-01027-f001:**
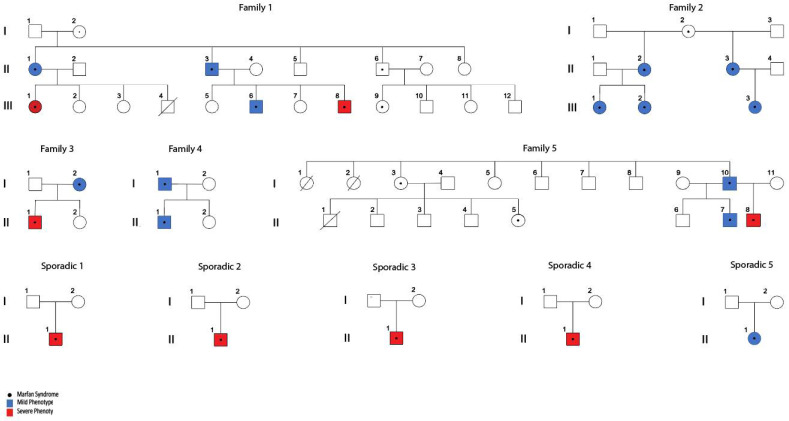
Familial and sporadic cases of MFS with significant variation in severity of their aortic phenotype. MFS clinical diagnosis was confirmed through identification of pathogenic variants in *FBN1* (see Table 1). Stratification of patients was done as described in Appendix A).

**Table 2 genes-13-01027-t002:** Number of candidates modifying genes identified under different VAAST analysis categories. Severe-Not_Mild: analysis to identify variants present in severe patients but not in mild patients. Mild-Not_Severe: analysis to identify variants present in mild patients but not in severe patients.

Samples	Mode of Inheritance	Analysis_ID	Candidate Genes (n)
**Fam1_all**	Dominant	1	7
	Recessive	2	8
**Fam 1.1**	Dominant	3	37
	Recessive	4	170
**Fam 1.2**	Dominant	5	14
	Recessive	6	31
**Fam 3**	Dominant	7	49
	Recessive	8	188
**Fam 5**	Dominant	9	5
	Recessive	10	6
**Severe-Not_Mild**	Dominant	11	12
	Recessive	12	15
**Mild-Not_Severe**	Dominant	13	95
	Recessive	14	97

**Table 3 genes-13-01027-t003:** Candidate modifying genes identified under different VAAST analysis categories.

Approach	Samples	Mode of Inheritance	Candidate Genes
**Familial**	**Fam 1.1**	Dominant	*PPARD*
		Recessive	*FBN1*
	**Fam 3**	Dominant	*MAP3K1*
		Recessive	*JAG1*
**Non-** **familial**	**Severe vs. Mild**	Dominant	*MAP3K1*, *PTPRJ*
		Recessive	*PTPRJ*
	**Mild vs. Severe**	Dominant	*KIAA1462*, *TNFSF18*, *TGFBR3L*
		Recessive	*TNFSF18*, *KIAA1462*

**Table 4 genes-13-01027-t004:** Genetic variants in candidate genes selected under different VAAST analysis categories.

Candidate Genes	Expression in Aorta	Genomic Coordinates (GRCh37/hg19)	Variants	MAF in gnomAD	Variant Type	SIFT	ACMG Classification
*PPARD*	Yes	Chr6:35387913	NM_001171818.2:c.140G>A	4.003 × 10^−5^	missense	tolerated-low confidence	Likely benign
*FBN1*	Yes	Chr15:48808467	NM_000138.5:c.1240C>A	N.R.	missense	tolerated	Likely pathogenic
*JAG1*	Yes	Chr20:10620449	NM_000214.3:c.3355G>C	N.R.	missense	tolerated	Likely pathogenic
		Chr20:10620450	NM_000214.3:c.3353T>A	N.R.	missense	tolerated	Uncertain significance
*MAP3K1*	Yes	Chr5:56111414	NM_005921.2:c.14C>G	4.225 × 10^−4^	missense	deleterious-low confidence	Uncertain significance
		Chr5:56177614	NM_005921.2:c.2587G>T	3.621 × 10^−5^	missense	tolerated-low confidence	Benign
		Chr5:.56111762	NM_005921.2:c.362G>A	N.R.	missense	tolerated	Uncertain significance
		Chr5:56168548-56168649	NM_005921.2:c.1504_1505+101del	N.R.	frameshift deletion	neutral	Pathogenic
*PTPRJ*	Yes	Chr11:48166437-48166511	NM_002843.4:c.2786_2786+73del	N.R.	frameshift deletion	unknown/ unassessed	Likely pathogenic
		Chr11:48002530-48002532	NM_002843.4:c.66_68del	N.R.	in-frame deletion	damaging	Uncertain significance
		Chr11:48161067G	NM_002843.4:c.2182G>A	1 × 10^−3^	missense	deleterious	Uncertain significance
*KIAA1462*	Yes	Chr10:30316501-30316503	NM_020848.4:c.2574_2576del	N.R.	frameshift deletion	unknown/ unassessed	Uncertain significance
		Chr10:30318653	NM_020848.4:c.424C>T	1 × 10^−^^3^	missense	tolerated	Likely benign
		Chr10:30316499-30316500	NM_020848.4:c.2577_2578insACTGCTGCT		in-frame insertion	unknown/ unassessed	Uncertain significance
*TNFSF18*	No	Chr17:173010746	NM_005092.4:c.361G>A	4.788 × 10^−5^	missense	tolerated	Uncertain significance
		Chr17:173010656	NM_005092.4:c.451A>G	3.989 × 10^−6^	missense	tolerated	Uncertain significance
		Chr17:173010651-173010652	NM_005092.4:c.455_456insTTG	N.R.	frameshift insertion	unknown/ unassessed	Uncertain significance
*TGFBR3L*	Yes	Chr19:7981648-7981650	NM_001195259.2:c.418_420del	N.R.	in-frame deletion	unknown/ unassessed	Uncertain significance

## Data Availability

Raw sequencing data and metadata is available under Bio Project accession number PRJNA795044 upon publication.

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
