# Peer review of "Exome Sequencing Identifies Genetic Variants Associated with Extreme Manifestations of the Cardiovascular Phenotype in Marfan Syndrome"

_genes, 2022, doi:10.3390/genes13061027_

Round 1

Reviewer 1 Report

Jimenez et al. submitted a research article entitled: Exome Sequencing Identifies Genetic Variants Associated with 2 Extreme Manifestations of The Cardiovascular Phenotype in 3 Marfan Syndrome.

The authors used WES to identify genetic variants associated with the severity of aortic manifestations and performed linkage analysis. They revealed 5 genes associated with severe aortic phenotype and 3 genes that could be protective for this phenotype in Marfan Syndrome (MFS).

 The data is exciting and could be of interest to the readers.

General comments:

  1. Either use mutation or variant. The term variant is preferred. Sequence Variant Nomenclature (hgvs.org). Kindly follow variant nomenclature.
  2. Kindly write the gene name in italic and the protein name in non-italic.
  3. Use a three-letter code for an amino acid.
  4. Write the (c. ) and (p. ) (NM_), (NP_)position for the variants.  Sequence Variant Nomenclature (hgvs.org).
  5. Add Sanger sequencing results for the 5 variants.
  6. Mention the frequency of the identified variants in public databases like gnomAD, and ExAC.
  7. It would be better to at least show 3D protein molding for the variant identified. To predict the effect of the identified variant on protein structure.
  1. A flowsheet of the Filtration steps and variant selection would help the readers.
  2. Classification of the identified variant according to ACMG should be mentioned.
  3. Better to move the pedigrees in the suppl file into the main text.

Reviewer 2 Report

This is a well written, interesting manuscript, that will contribute to the literature.

Author Response

Thanks for your comments.

This manuscript is a resubmission of an earlier submission. The following is a list of the peer review reports and author responses from that submission.

Round 1

Reviewer 1 Report

Marfan syndrome (MFS) is caused by errors in FBN1 gene (the primary mutation). This manuscript described the identification of MFS genetic modifiers by whole-exome sequencing from 5 families and 5 non-familial cases. It should be noted that even different mutations in FBN1 gene also have significant effects on disease severity (Arnaud P, et al. 2021. doi:10.1038/s41436-021-01132-x). The observed variability in severity is contributed by primary mutations, non-heritable environmental factors, and other genetic modifiers (primary mutation > environmental factors ? other genetic modifiers). Based on abdominal aortic aneurysm heritability estimation (e.g., Joergensen TMM et al, 2016. doi:10.1016/j.ejvs.2016.03.012) and other reports (e.g., smoking is believed to be a risk factor for MFS patients), I tend to believe environmental factors matter more than other genetic modifiers. This might not be true becasue identification of genetic modifiers is difficult (large sample size or family size, or additional molecular evidence is required. Aubart M et al identified 1070 clinically well-characterized FBN1 disease-causing variant carriers and identified several genetic modifiers for MFS (2018 Eur J Hum Genet. This is the most related paper but I do not know why it is not cited here). For other rare genetic disorders, especially those with late-onset subtypes (e.g., Pompe disease caused by errors in GAA gene), the contribution of primary mutations, non-heritable environmental factors, and other genetic modifiers are similar. 

The authors compared variants between mild and severe cases to identify genetic modifiers. Practically they applied two strategies. 
First, they compared mild and severe cases in families 1,3 and 5 (family 1: N=3 mild vs 2 severe, family 3: 1 vs 1, and family 5: 2 vs 1). In this scenario, the primary mutations were assumed the same in each family (with WES data, this can be tested; I do not see a clear statement in the manuscript). Therefore the variability was caused by genetic modifiers and environmental factors, which is important but the authors ignore. Even though the genetic modifier exists here, the small sample size (the largest is 3 vs 2 in family 1) might not provide enough statistical power to identify it. The authors use VAAST to compute likelihood p-values and consider those adjusted p<0.05 to be candidate genes, then select those with disease relevance (MFS GO term, TGF pathway and cardiovascular phenotypes). Two problems here. First, VAAST are not designed to be used directly to identify causal variant using p or adjust p of 0.05 (even the developer did not use it in this way). The reported p-value does not reflect the real false discovery rate. It is better to generate a pool using fixed score cutoff. Second, because of the use of disease relevance information, the identified genes will always look good. There will be no way to control the false positives, which might be high in this study. The authors selected 4 variants from a pool of several hundreds of variants (the pool itself could be not reliable and easily changed due to parameters). These four genes sound good, but it is unclear whether the selected four variants affect gene functionality. 

Second, they conducted association analysis combining all families using the “incomplete penetrance” parameter. Given the small sample size, the lack of variates controlled for the primary causal variant (*very important*), and the lack of kinship, the identified variants are not trustable without additional evidence. A detailed description of the primary variants is also required. 

The analysis could still be valuable if the authors provide additional molecular evidence experimentally or computationally. For example, the authors could test if gene expression is altered by these variants (e.g., eQTL analysis). The authors could use the identified variants to construct a pool for a genetic screen towards disease-related assays. 

In addition, the authors should reveal more technical details/results to convince the reader. For each of the selected variants, generate a figure to show how are they are distributed in each family. 

Reviewer 2 Report

The introduction section is weak. The novelty of the work has not been explained well. There is no link between the introduction and results sections. It is a bit confusing for me to understand the importance of the current work. What makes this work different from the previous published paper. The english needs to massively improved.